Subject Area:
immunology/cellular biology

Keywords:
mir-149-3p, PD-1, T-cell exhaustion, breast cancer, immunotherapy

Authors for correspondence:
Dian Gao
e-mail: gaodian@ncu.edu.cn
Weiping Min
e-mail: weiping.min@uwo.ca

# miR-149-3p reverses CD8+ T-cell exhaustion by reducing inhibitory receptors and promoting cytokine secretion in breast cancer cells

Meng Zhang[1,3,4,5,6], Dian Gao[2], Yanmei Shi[1,3,4,5,6,7], Yifan Wang[1,3,4,5,6], Rakesh Joshi[3,4,5,6], Qiongfang Yu[8,9], Daheng Liu[3,4,5,6], Faizah Alotaibi[3,4,5,6], Yujuan Zhang[1,2], Hongmei Wang[1], Qing Li[8,9], Zhu-Xu Zhang[1,3,4,5,6], James Koropatnick[3,4,5,6] and Weiping Min[1,3,4,5,6,7]

[1]Institute of Immunotherapy of Nanchang University, and Jiangxi Academy of Medical Sciences, Nanchang 330006, People's Republic of China
[2]Department of Pathogen Biology and Immunology, Medical College of Nanchang University, Nanchang 330006, People's Republic of China
[3]Department of Surgery, [4]Department of Pathology and Laboratory Medicine, [5]Department of Oncology, and [6]Department of Microbiology and Immunology, Western University, London, Canada N6A 5A5
[7]Department of Oncology, the First Affiliated Hospital of Nanchang University, Nanchang 330006, People's Republic of China
[8]Department of Gastroenterology and Hepatology, and [9]Department of Oncology, Second Affiliated Hospital of Nanchang University, Nanchang 330006, People's Republic of China

iD WM, 0000-0001-7389-3194

Blockade of inhibitory receptors (IRs) is one of the most effective immunotherapeutic approaches to treat cancer. Dysfunction of miRNAs is a major cause of aberrant expression of IRs and contributes to the immune escape of cancer cells. How miRNAs regulate immune checkpoint proteins in breast cancer remains largely unknown. In this study, downregulation of miRNAs was observed in PD-1-overexpressing CD8+ T cells using miRNA array analysis of mouse breast cancer homografts. The data reveal that miR-149-3p was predicted to bind the 3'UTRs of mRNAs encoding T-cell inhibitor receptors PD-1, TIM-3, BTLA and Foxp1. Treatment of CD8+ T cells with an miR-149-3p mimic reduced apoptosis, attenuated changes in mRNA markers of T-cell exhaustion and downregulated mRNAs encoding PD-1, TIM-3, BTLA and Foxp1. On the other hand, T-cell proliferation and secretion of effector cytokines indicative of increased T-cell activation (IL-2, TNF-α, IFN-γ) were upregulated after miR-149-3p mimic treatment. Moreover, the treatment with a miR-149-3p mimic promoted the capacity of CD8+ T cells to kill targeted 4T1 mouse breast tumour cells. Collectively, these data show that miR-149-3p can reverse CD8+ T-cell exhaustion and reveal it to be a potential antitumour immunotherapeutic agent in breast cancer.

## 1. Introduction

T-cell exhaustion was first defined as a state of immune dysfunction in chronic lymphocytic choriomeningitis virus infection [1]. Growing evidence reveals that exhausted T cells are widely distributed in virus-infected tissues and the tumour microenvironment (TME) [2,3]. During the process of exhaustion, T cells chronically exposed to tumour antigens or viral antigens gradually lose their ability to kill cancer cells or virus-infected cells expressing those antigens. Exhaustion is characterized by elevated levels of inhibitory receptors (IRs) such as programmed death-1 (PD1), cytotoxic T-lymphocyte antigen 4 (CTLA4),

royalsocietypublishing.org/journal/rsob    Open Biol. 9: 190061

T-cell immunoglobulin domain and mucin domain 3 (TIM3), B- and T-lymphocyte attenuator (BTLA), and lymphocyte activation gene 3 (LAG3), CD244 (2B4) [4–9]; diminished levels of effector cytokines, such as interlekin-2 (IL-2), tumour necrosis factor-α (TNF-α) and interferon-γ (IFN-γ); and impaired CD8$^+$ T-cell cytotoxicity [10]. Foxp1, a member of subfamily P of the forkhead box (FOX) transcription factor family, plays a pivotal role in modulating early B-cell development, T-cell quiescence and monocyte differentiation, and in mediating effective antitumour immune response [11–14]. Foxp1 is highly expressed in oestrogen receptor-positive human breast cancer cells and can inhibit the migration of tumour-infiltrating T cells [15]. Moreover, Foxp1 is overexpressed in tumour-infiltrating lymphocytes [16]. Targeting Foxp1 may improve PD-1/PD-L1 pathway-associated antitumour immunity.

MicroRNAs (miRNAs) are a family of non-coding RNAs (approx. 21 nucleotides in length) [17,18]. They are widely expressed by plant, animal and viral genomes, and participate in post-transcriptional regulation of many genes by binding the 3' untranslated regions (3'UTRs) of mRNAs to target them for degradation and/or inhibition of translation [19,20]. Aberrant expression of miRNAs is involved in the pathogenesis of various diseases including cancer, cardiac hypertrophy, respiratory diseases and others [21–24]. Depending on the circumstances, miRNAs can either promote or suppress tumour formation by modulating cell proliferation, death, invasion, metastasis and/or angiogenesis [25–27]. In immune cells within the TME, miRNAs can stimulate or suppress antitumour immunity by controlling immune regulatory molecules in both tumours and immune cells [28]. Multiple miRNAs have been identified as regulators of immune escape by directly or indirectly modulating the expression of immune-regulating molecules, especially immune checkpoint proteins such as members of the CD28, B7, TNF and TNFR families [29].

Breast cancer is the most common cause of cancer death in women [30]. Currently, the regulation of IRs is a promising method to treat breast cancer [31]. Aberrant miRNA expression, leading to post-transcriptional induction of pathological IR expression and activity, may lead to cancer cell escape from immune surveillance. Recently, we and other researchers have reported that miR-28, miR-138, miR-4717 and miR374b can regulate the expression of PD-1 and influence the immune status of T cells in some cancers [32–35], revealing the modulatory roles of miRNAs in either reducing or enhancing T-cell function. However, to date, studies have focused primarily on how miRNAs regulate CD4+ T cells or cytokine-induced killer (CIK) cells. In breast cancer, it is not clear how miRNAs regulate IRs and PD-1-associated transcriptional factor Foxp1 in CD8$^+$ T cells. Therefore, we have focused on miRNAs targeting iR and Foxp1 gene expression in CD8$^+$ T cells in murine 4T1 breast cancer cells *in vitro*; our data reveal that an miRNA has the capacity to enhance antitumour immunity by reversing CD8$^+$ T-cell exhaustion.

# 2. Results

## 2.1. Overexpression of IRs in spleen CD8$^+$ T cells from 4T1 tumour-bearing mice

To evaluate CD8$^+$ T-cell exhaustion in 4T1 breast tumour-bearing mice, we examined the level of IR mRNAs, including PD-1, TIM-3, BTLA and exhausted T-cell-associated transcriptional factor Foxp1 using reverse transcription quantitative polymerase chain reaction (RT–qPCR). Compared to those of naive mice spleen CD8$^+$ T cells, the levels of PD-1, TIM-3, BTLA and Foxp1 were upregulated in tumour-bearing mice (figure 1a). We next evaluated the levels of T-cell-exhausted phenotype markers on CD8$^+$ T cells by flow cytometry. Population of CD8$^+$ T cells was decreased in tumour-bearing mice (electronic supplementary material, figure S1A,B). Compared to that of controls, the percentage of PD-1+ cells among CD8$^+$ T cell was increased from 14.6 to 21.6% ($p = 0.019$) in 4T1 tumour-bearing mice. Furthermore, the percentage of TIM-3+ cells among CD8$^+$ T cells was increased from 12.6 to 22% ($p = 0.011$). There was no apparent difference in the ratio of BTLA+ cells to CD8$^+$ T cells between the two groups (figure 1b).

## 2.2. Downregulation of cytokine secretion in CD8$^+$ T cells isolated from spleens of tumour-bearing mice

To assess the cytotoxicity of CD8$^+$ T cells from spleens of 4T1-bearing mice, mixed lymphocyte reactions (MLRs) were performed. Lymphocytes from 4T1 tumour-bearing mice and naive mouse spleens were co-cultured with C57BL/6 bone marrow-derived dendritic cells (DCs) for 48 h. Cytokine receptor levels were then assessed by flow cytometry. The fraction of CD8$^+$ T cells (IL-2+, TNFα+ or IFN-γ+) decreased in CD8$^+$ T cells from 4T1-bearing mouse spleens compared with CD8$^+$ T cells from spleens of tumour-naive mice (figure 2a–f).

## 2.3. Decreased CD8$^+$ T-cell response in tumour-bearing mice

To determine the homeostatic proliferation/differentiation of CD8$^+$ T cells, a CFSE dye dilution assay of proliferation was conducted. The proliferation of CD8$^+$ T cells declined in tumour-bearing mice on day 3 (figure 3a).

To detect the survival of CD8$^+$ T cells, we examined the ratio of apoptosis in lymphocytes from naive mice to apoptosis in CD8$^+$ T cells from spleens of tumour-bearing mice (the apoptosis ratio). Annexin V and PI staining showed that the apoptosis ratio increased from 19.9 to 27.7% ($p = 0.042$) in CD8$^+$ T cells from tumour-bearing mice (figure 3b).

## 2.4. Global miRNA level profile and miRNAs which can target IRs and Foxp1

CD8$^+$PD-1+ and CD8$^+$PD-1− T cells were isolated from spleens of 4T1 breast tumour-bearing mice by flow cytometric cell sorting. Affymetrix GeneChip 3.0 miRNA array analysis was performed on RNAs extracted from each of the two groups. Probe-level data were generated using Affymetrix Command Console v. 3.2.4. Probes were summarized to the miRNA level using RMA. Partek was used to determine ANOVA $p$-values and fold changes for miRNAs. miRNAs with a fold change of ±1.5 (PD-1+ versus PD-1−, $p < 0.05$) that were screened are shown in a heat map as candidate miRNAs (figure 4a). Multiple databases, including miRWalk, Targetscan, miRanda and others, were used to predict the binding sites of candidate miRNAs. Among them, five miRNAs (miR-149-3p, miR-146a-3p, miR-122-5p, miR-211-5p

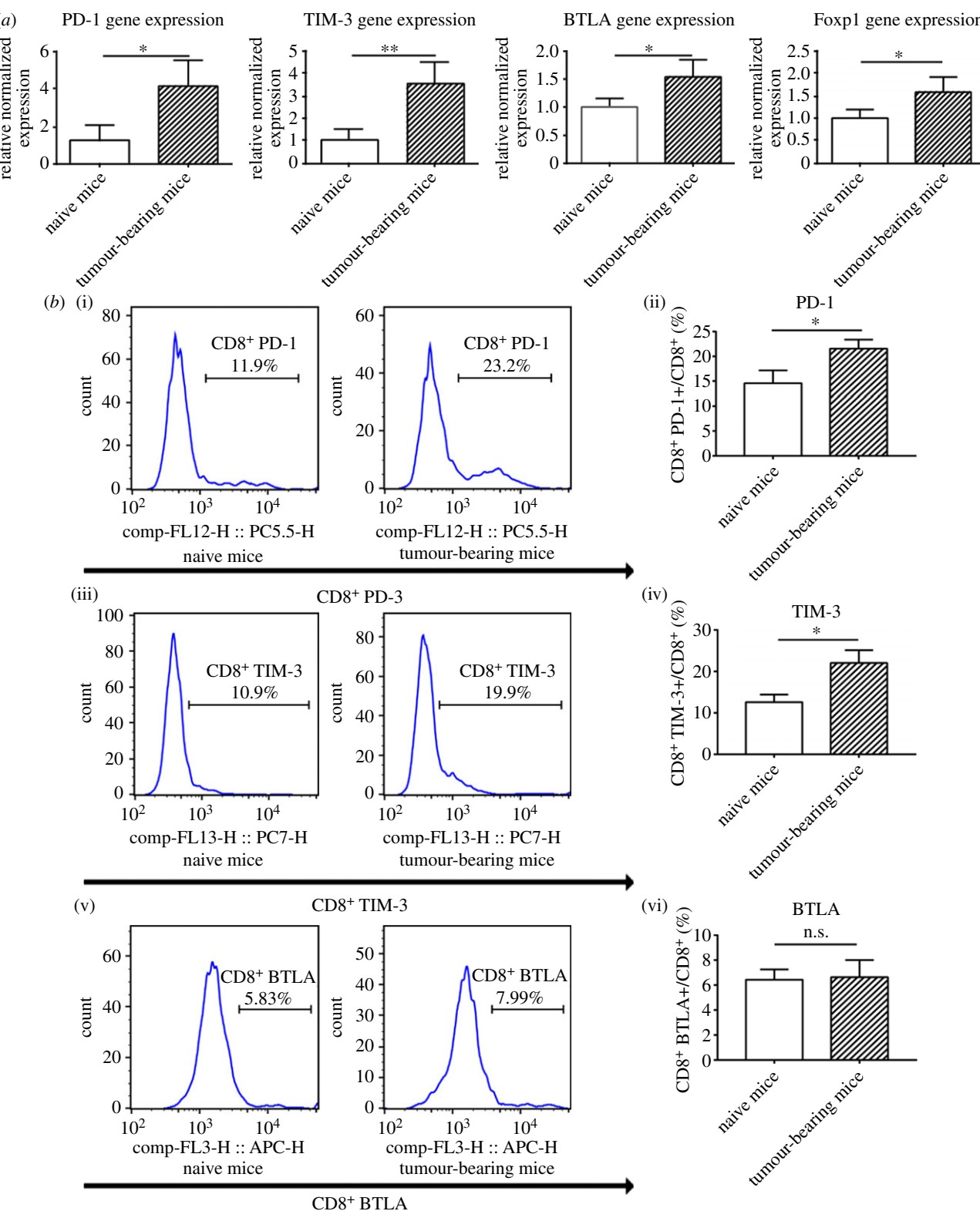

**Figure 1.** Exhaustion marker upregulation in CD8⁺ T cells from spleens of 4T1 tumour-bearing mice. (*a*) Detection of mRNAs encoding PD-1, TIM-3, BTLA by RT–qPCR. Spleen cells were collected from naive mice and from tumour-bearing mice on days 16–18 after tumour cell injection. CD8⁺ T cells were purified from the collected splenocytes using Miltenyi magnetically labelled beads (Miltenyi Biotec). RT–qPCR was performed to detect PD-1, TIM-3, BTLA and Foxp1 mRNA levels in CD8⁺ T cells. (*b*) Detection of IRs on CD8⁺ T cells by flow cytometry. Spleen cells were collected from naive mice and from tumour-bearing mice. Flow cytometry was performed to detect PD-1, TIM-3 and BTLA expression level. Data are representative of three independent experiments. Unpaired Student's *t*-tests were performed to determine statistical significance (*$p < 0.05$, **$p < 0.01$).

and miR-721) with sequences predicted to bind the 3′UTR of IRs and/or TF genes were further examined (figure 4*b*). miR-149-3p, complementary to the 3′UTRs of mRNAs encoding PD-1, TIM-3, BTLA and Foxp1, was further confirmed to be downregulated in tumour-bearing mouse spleens (figure 4*c*).

## 2.5. miR-149-3p downregulated exhausted T-cell phenotype *in vitro*

When CD8⁺ T cells from spleens of 4T1 breast tumour-bearing mice were transfected with miR-149-3p mimic for 48 h, the level of miR-149-3p in CD8⁺ T cells was upregulated

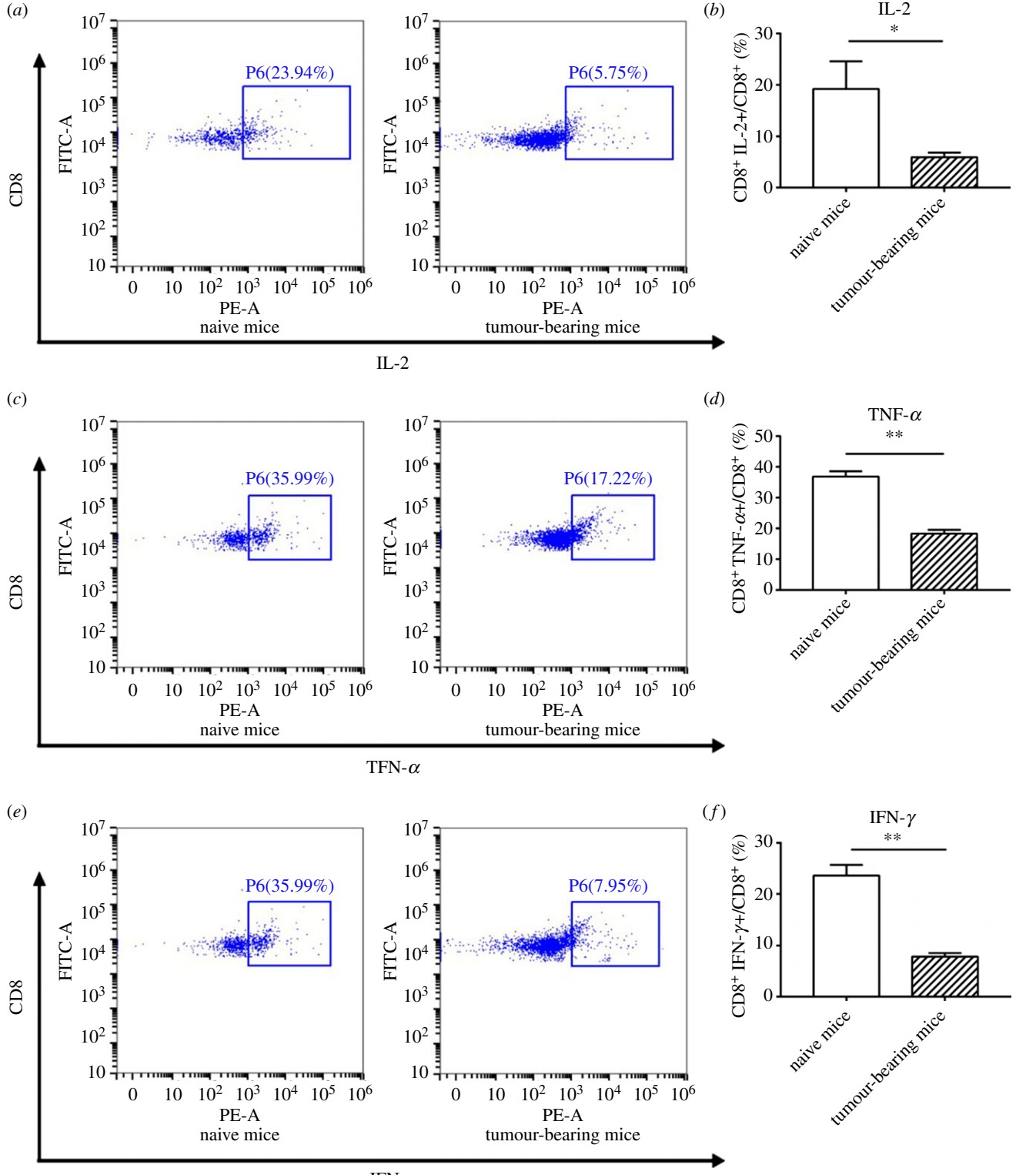

**Figure 2.** Downregulation of cytokine secretion in CD8+ T cells isolated from spleens of mice-bearing 4T1 tumours. (*a–f*) The fraction of IL-2+, TNF-α+ and IFN-γ+ CD8+ T cells was measured by flow cytometry. Spleen cells were collected from naive mice and tumour-bearing mice and co-cultured with C57BL/6 bone marrow-derived DCs for 48 h. Flow cytometry was performed to detect IL-2, TNF-α and IFN-γ expression level. Unpaired Student *t*-test analysis was performed to determine statistical significance (*$p < 0.05$, **$p < 0.01$).

after miR-149-3p mimic transfection (electronic supplementary material, figure S2). The levels of mRNAs encoding PD-1, TIM-3, BTLA and Foxp1 decreased (figure 5*a*). Conversely, the level of mRNAs encoding these genes was increased when inhibitors of miR-149-3p were used (figure 5*a*).

The function of miR-149-3p in regulating the exhausted T-cell phenotype was also assessed by a flow cytometric analysis. Forty-eight hours after miR-149-3p mimic transfection of

CD8+ T cells isolated from spleens of 4T1 tumour-bearing mice, the population of PD-1+ CD8+ T cells decreased from 34.7% to 26.8% ($p = 0.005$). Moreover, the population of TIM-3+ CD8+ T cells declined from 27.5% to 23.7% ($p = 0.031$) and the population of BTLA+ CD8+ T cells was downregulated from 13.8% to 9.0% ($p = 0.006$) (figure 5*b*). Conversely, when miR-149-3p inhibitors were used, the population of BTLA+ CD8+ T cells increased from 13.8% to 16.8% ($p = $

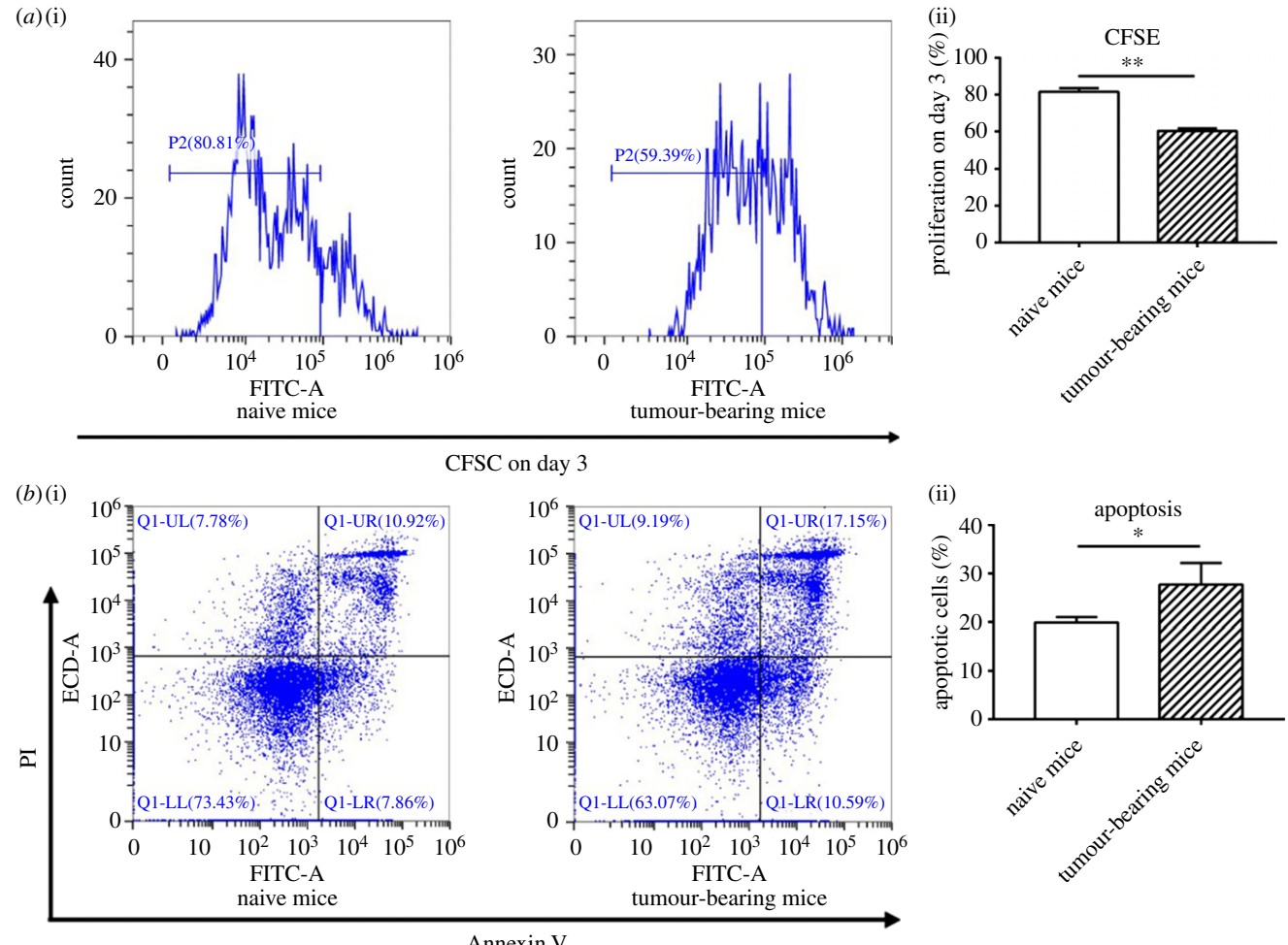

**Figure 3.** CD8$^+$ T cells isolated from spleens of tumour-bearing mice proliferated less and were more apoptotic (indicators of reduced T-cell function). (a) Proliferation was detected by CFSE staining on day 3. Spleen cells were collected from naive mice, and from tumour-bearing mice on days 16–18 after tumour cell injection. CD8$^+$ T cells were purified from the collected splenocytes using Miltenyi magnetically labelled beads. T cells were labelled with CFSE before co-culture with C57BL/6 bone marrow-derived DCs. Proliferation analysis, by flow cytometry, was performed on day 3. (b) Apoptosis was detected by Annexin V/PI staining. Spleen cells were collected from naive mice and tumour-bearing mice. CD8$^+$ T cells were purified from the collected splenocytes using Miltenyi magnetically labelled beads. Apoptosis was detected by flow cytometry within 1 h of FITC-Annexin-V and PI staining. Data are representative of three independent experiments. Unpaired Student's $t$-test analyses were performed to determine statistical significance (*$p < 0.05$, **$p < 0.01$).

0.045). There was no significant change in the population of PD-1+ and TIM-3+ CD8$^+$ T cells (figure 5$b$).

## 2.6. miR-149-3p restored activity-associated cytokine levels in exhausted CD8$^+$ T cells

We further examined the effect of miR-149-3p on the level of activity-associated cytokines and proliferation of CD8$^+$ T cells isolated from spleens of 4T1 tumour-bearing mice. After transfection with miR-149-3p mimic or inhibitor, T cells were co-cultured with C57BL/6 bone marrow-derived DCs. After treatment with miR-149-3p mimic, the population of CD8$^+$/IL-2+ T cells among all CD8$^+$ T cells increased from 15.7% to 28.3% ($p = 0.001$) (figure 6$a$,$b$). Similarly, the population of CD8$^+$/TNF-α+ among all CD8$^+$ T cells increased from 24.1% to 33.5% ($p = 0.010$) (figure 6$c$,$d$). In addition, the population of CD8$^+$/INF-γ+ T cells among all CD8$^+$ T cells increased from 31.8% to 36.7% ($p = 0.022$) (figure 6$e$,$f$). CD8$^+$/IL-2+ T cells decreased from 15.7% to 10.1% ($p = 0.043$) (figure 6$a$,$b$) and CD8$^+$/TNF-α+ T cells decreased from 24.1% to 17% ($p = 0.030$) (figure 6$c$,$d$) after treatment with a miR-149-3p inhibitor. To further verify changes in

cytokine levels, we repeated our experiments and detected cytokine mRNA levels by qPCR. After treatment with an miR-149-3p mimic, IL-2, TNF-α and IFN-γ mRNA levels were upregulated, whereas IL-2, TNF-α and IFN-γ mRNAs were downregulated after treatment with a miR-149-3p inhibitor (electronic supplementary material, figure S3).

## 2.7. miR-149-3p mimic transfection increased proliferation and decreased apoptosis in exhausted CD8$^+$ T cells

After transfection with miR-149-3p mimic or inhibitor, spleen CD8$^+$ T cells from 4T1 tumour-bearing mice were co-cultured with C57BL/6 bone marrow-derived DCs from mice without 4T1 tumour homografts. CD8$^+$ T cells treated with miR-149-3p mimic displayed increased proliferation, while proliferation decreased when CD8$^+$ T cells were transfected with miR-149-3p inhibitor (figure 7$a$).

In addition, the percentage of apoptotic CD8$^+$ T cells decreased from 50.7% to 45.2% ($p = 0.008$) after the cells were transfected with miR-149-3p mimic for 48 h (figure 7$b$).

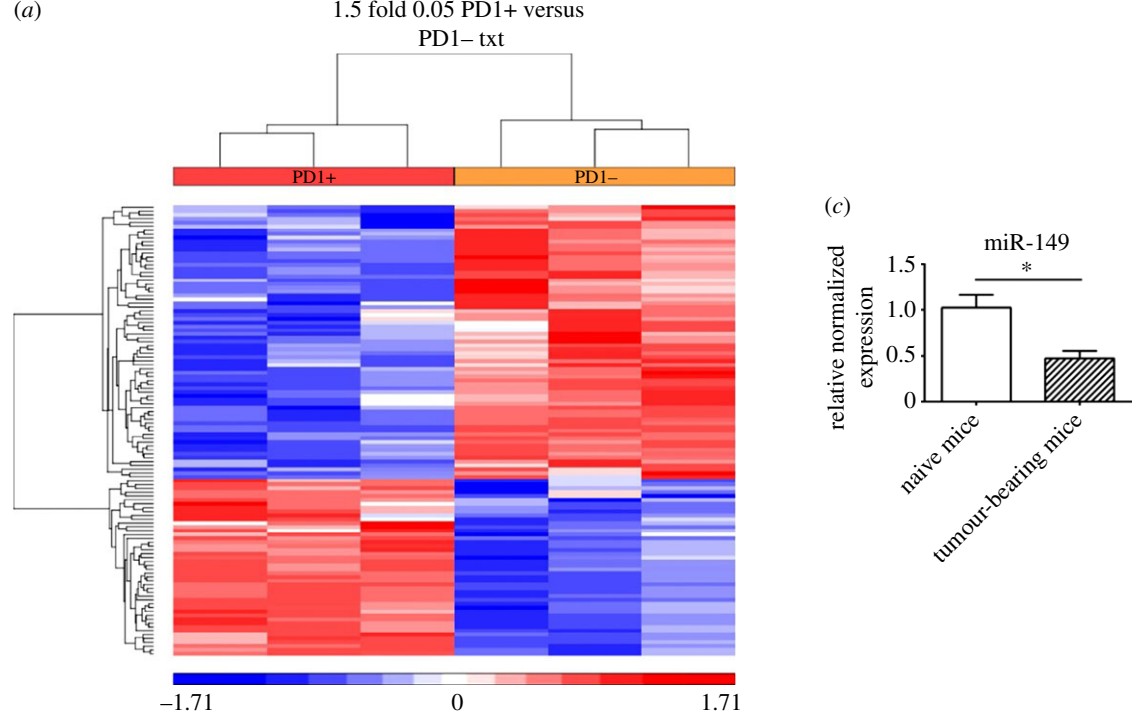

**Figure 4.** Global miRNA level profile analysis and potential targeting of exhaustion-associated IR mRNAs by miR-149-3p. (*a*) Heat map of miRNAs with differential levels between CD8$^+$PD1− and CD8$^+$PD1+ samples. Total RNA was labelled using a Flash Tag Biotin HSR kit (Genisphere) and hybridized to Affymetrix miRNA 3.0 arrays. A total of 117 miRNAs with significant changes were found (electronic supplementary material, table S1). (*b*) Predicting miRNAs that target IR mRNAs by miRNA walk, Targetscan, miRanda and other databases. Among 117 miRNAs, 5 candidate miRNAs, including miR-149-3p, miR-146a-3p, miR-122-5p, miR-211-5p and miR-721, were selected on the basis of their potential to bind the 3′UTRs of mRNAs encoding PD-1, TIM-3, BTLA and Foxp1. (*c*) miR-149-3p levels were confirmed by RT–qPCR. Total RNA generated from naive and tumour-bearing mice was used to detect the level of miR-149-3p. Data are representative of three independent experiments. Unpaired Student's *t*-test analyses were performed to determine statistical significance (*$p < 0.05$).

The fraction of apoptotic CD8$^+$ T cells was not altered by treatment with miR-149-3p inhibitor.

## 2.8. miR-149-3p mimic treatment promotes cytotoxic CD8$^+$ T-cell killing of mouse 4T1 tumour cells *in vitro*

To examine whether miR-149-3p could affect the capacity of T cells to kill mouse 4T1 breast tumour cells, the capacity of cytotoxic CD8$^+$ T cells to induce the death of 4T1 in co-culture was assessed. Tumour cytotoxicity of CD8$^+$ T cells improved when CD8$^+$ T cells were treated with miR-149-3p mimic (figure 8).

## 3. Discussion

Immune checkpoint blockade, which enhances T-cell activation and/or T-cell survival, has resulted in remarkable outcomes in anti-cancer immunotherapy. However, specific monoclonal antibodies directed against specific inhibitor receptors suppress single molecules rather than multiple targets included within

regulons (collections of molecules mediating whole regulatory pathways and complex physiological events). The use of monoclonal antibodies therefore limits the potential for combinatorial expansion for therapeutic targeting of whole physiological pathways a challenge in the clinic [36]. One specific miRNA can modulate the expression of several genes, making miRNA-based immunotherapeutics a potential new and effective approach in combinatorial anti-cancer therapy. A growing number of studies have confirmed that miRNA-IR regulatory axes play a critical role in immune escape and immune checkpoint therapy [29]. Our current study finds that miRNA-149-3p, identified by screening and assessing multiple miRNA profiles, potentially interacts with inhibitory T-cell receptors PD-1, Tim3, BTLA and PD-1-associated transcriptional factor Foxp1, and exerts potentially anti-cancer efficacy by reversing CD8$^+$ T-cell exhaustion. Reversal of T-cell exhaustion is critical in promoting cytotoxic T-cell-mediated antitumour immunity, and these data support the possibility of miRNA-based immunotherapy of breast cancers.

In previous studies, T-cell exhaustion, leading to dysfunction and reduced antitumour immune response, was observed in clinical human breast cancers and mouse models [37,38]. However, the regulatory processes mediating CD8$^+$ T-cell

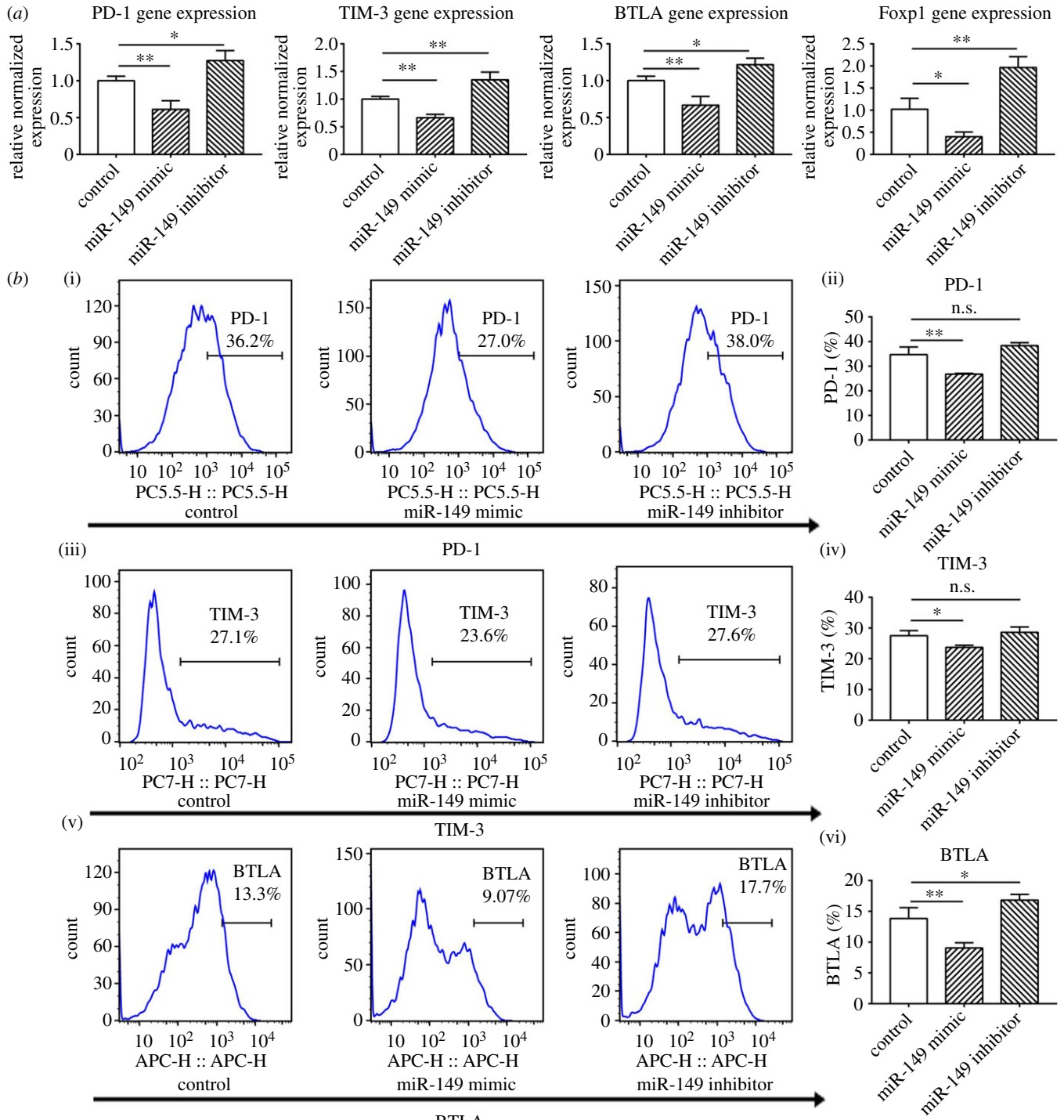

**Figure 5.** Altered expression of markers of exhaustion in T cells after treatment with miR-149-3p mimic or miR-149-3p inhibitor. (a) Detection of mRNAs encoding PD-1, TIM-3, BTLA and Foxp1 by RT–qPCR analysis. CD8+ T cells were purified from the tumour-bearing mice splenocytes using Miltenyi magnetically labelled beads and transfected with control miRNA, miR-149-3p mimics or miR-149-3p inhibitors for 48 h. After transfection, RT–qPCR was performed to detect PD-1, TIM-3, BTLA and Foxp1 mRNA levels in CD8+ T cells. (b) Examination of IRs on CD8+ T cells by flow cytometry. CD8+ T cells were purified from tumour-bearing mice splenocytes and transfected with control miRNA, miR-149-3p mimics or miR-149-3p inhibitors for 48 h. Flow cytometry was performed after transfection to detect PD-1, TIM-3 and BTLA expression level. Data are representative of three independent experiments. One-way ANOVA analyses were performed to determine statistical significance ($*p < 0.05$, $**p < 0.01$).

exhaustion remain largely unknown. Consistent with the overexpression of IRs in human melanoma, lymphoma, hepatocellular carcinoma and gastric cancer [39–41], our results show that mRNAs encoding PD-1, TIM-3, BTLA and Foxp1 are also upregulated in CD8+ T cells isolated from spleens of mice-bearing mouse 4T1 breast tumours. In addition, T-cell-exhausted phenotypes of the above IRs reveal a similar upregulating model (with the exception of BTLA) compared with that from naive mice. Reduced levels of activation-associated

cytokines (IL-2, TNF-α, IFN-γ) are an important characteristic of exhausted T cells [42,43], and we report that the levels of these cytokines in CD8+ T cells isolated from spleens of mice-bearing mouse 4T1 breast tumours are, as expected, similarly reduced. Thus, markers of T-cell exhaustion in splenic CD8+ T cells from mice-bearing 4T1 tumour homografts follow a pattern, indicating that this is a useful model of T-cell exhaustion and resulting dysfunction relevant to antitumour immune surveillance.

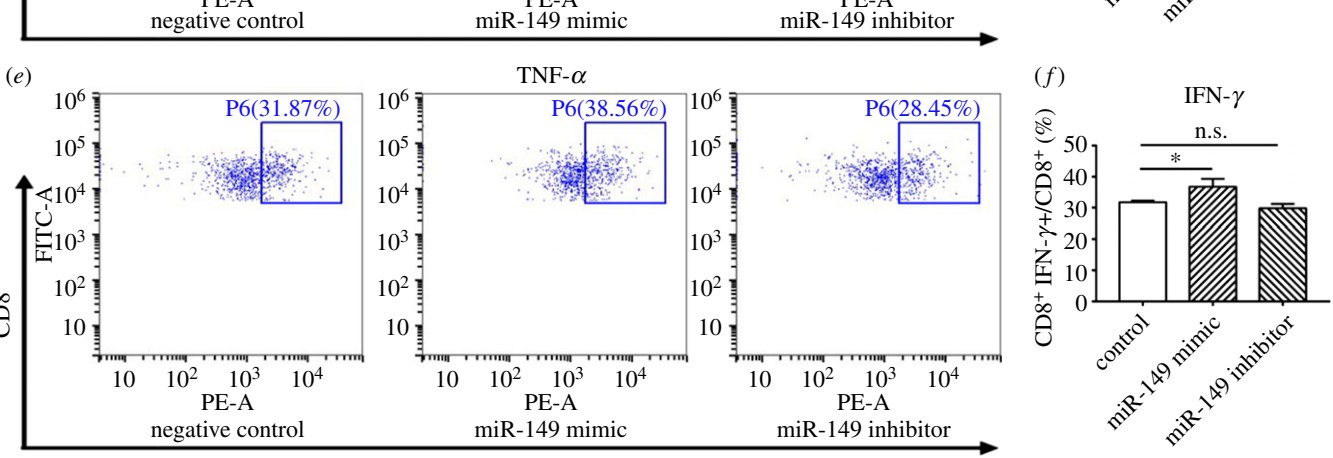

**Figure 6.** miR-149-3p modulation in T cells regulates cytokine production. (a–f) Cytokines were detected by flow cytometry. Spleen cells collected tumour-bearing mice transfected with control miRNA, miR-149-3p mimics or miR-149-3p inhibitors for 48 h and co-cultured with C57BL/6 bone marrow-derived DCs for 48 h. Flow cytometry was performed to detect IL-2, TNF-α and IFN-γ expression level. One-way analyses of variance were performed to determine statistical significance (*$p < 0.05$, **$p < 0.01$).

Co-culture of spleen CD8$^+$ T cells from mice-bearing 4T1 tumour homografts with bone marrow-derived DCs from naive mice without tumour homografts decreased CD8$^+$ T-cell proliferation in 4T1 breast tumour-bearing mice spleen CD8$^+$ T cells, compared with naive mice. Apoptosis was increased in CD8$^+$ T cells from tumour-bearing mouse spleens. Both these observations reveal decreased immune response manifested by decreased effector cytokine levels and increased T-cell apoptosis in the presence of 4T1 breast tumour homografts.

Blocking IRs has been recognized as an important potential strategy to improve antitumour immune response [44]. To determine which miRNAs are possible targets for therapeutic targeting, we compared miRNA levels from CD8$^+$PD1+ (exhausted) and CD8$^+$PD-1− (non-exhausted) T cells isolated from mice-bearing 4T1 breast tumours. According to miRNA array data and RT–qPCR verification, the level of miR-149-3p was confirmed to be decreased in CD8$^+$PD-1− T cells. Multiple databases further predicted that miR-149-3p had

binding sites which target to 3′UTRs of mRNAs encoding PD-1, TIM-3, BTLA and TF Foxp1. miR-149-3p, therefore, has a potential to restore activity to exhausted T cells by reducing IR levels in CD8$^+$ T cells.

Previous reports by us and others show that miR-28, miR-138, miR-4717 and miR374b can directly target PD-1 and partly restore the function of exhausted T cells in cancers [32–35]. In this study, the application of an miR-149-3p mimic reduced the levels of IRs associated with T-cell exhaustion (PD-1, TIM-3, BTLA and Foxp1) in CD8$^+$ T cells, and reduced proliferation and apoptosis: phenotypic markers of T-cell exhaustion. These data suggest that the use of the miR-149-3p mimic restored T-cell function by reducing T-cell markers of exhaustion and increasing T-cell markers of activation. Furthermore, the percentage of PD-1+ CD8$^+$ T cells, TIM-3+ CD8$^+$ T cells and BTLA+ CD8$^+$ T cells among the total population of all CD8$^+$ T cells was decreased in T cells treated with a miR-149-3p mimic. Thus, miR-149-3p can reduce the levels of IRs in CD8$^+$ exhausted T cells.

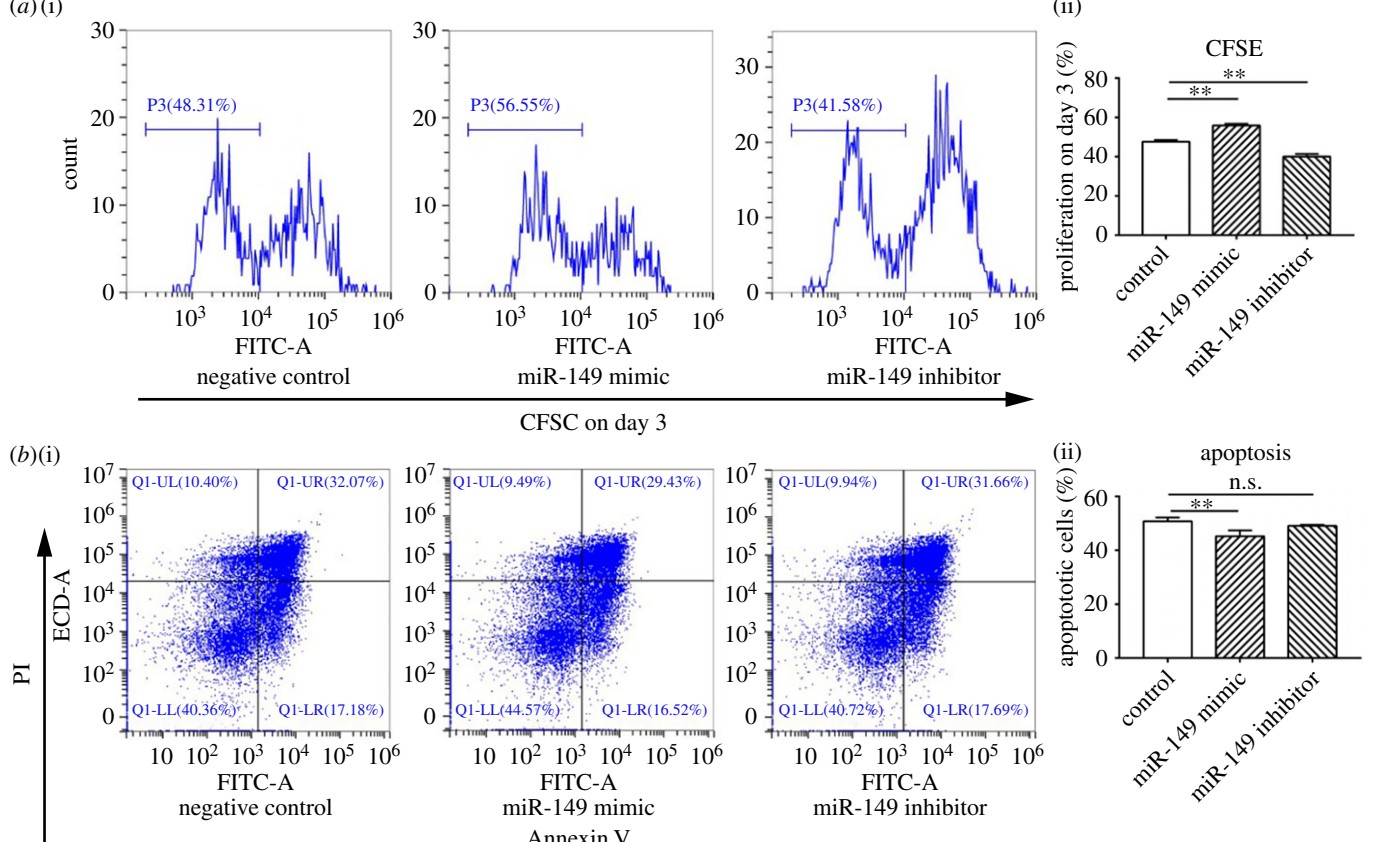

**Figure 7.** miR-149-3p mimic transfection restores exhausted CD8+ T-cell functions *in vitro*. (*a*) Proliferation was detected on day 3 by CFSE staining. Spleen cells collected from tumour-bearing mice transfected with control miRNA, miR-149-3p mimics or miR-149-3p inhibitors for 48 h. CD8+ T cells were purified from the collected splenocytes using Miltenyi magnetically labelled beads. T cells were labelled with CFSE and co-cultured with C57BL/6 bone marrow-derived DCs. Proliferation was assessed by flow cytometry on day 3. (*b*) Apoptosis was detected by Annexin V/PI staining. Spleen cells were collected from tumour-bearing mice transfected with control miRNA, miR-149-3p mimics or miR-149-3p inhibitors for 48 h.CD8+ T cells were purified from the collected splenocytes using Miltenyi magnetically labelled beads. Apoptosis was detected by flow cytometry within 1 h after Annexin V-FITC and PI staining. One-way ANOVA analyses were performed to determine statistical significance (*$p < 0.05$, **$p < 0.01$).

Downregulated levels of cytokines associated with T-cell activation are another characteristic of exhausted T cells in the presence of tumours. Therefore, promoting the production of cytokines can effectively elevate CD8+ T-cell immunity response against tumours [45]. In this study, IL-2, TNF-α and IFN-γ levels were increased in exhausted CD8+ T cells after miR-149-3p mimic transfection, indicating that miR-149-3p can upregulate cytokine level secretion to potentially reverse CD8+ T-cell exhaustion.

Furthermore, after an miR-149-3p mimic was transfected into CD8+ T cells, proliferation was increased and apoptosis was decreased. These data suggest that miR-149-3p restored exhausted T-cell immune function in the presence of exhaustion-promoting tumours. In addition, we evaluated whether miR-149-3p could modulate CD8+ T-cell cytotoxicity against 4T1 breast tumour cells. The cytotoxicity of CD8+ T cells from tumour-bearing mice was upregulated after transfection with the miR-149-3p mimic, suggesting that miR-149-3p has the potential to enhance CD8+ T-cell antitumour immunity.

# 4. Conclusion

Our results show that miR-149-3p has the potential to target the IRs PD-1, TIM-3 and BTLA. It can promote CD8+ T-cell-mediated immune response and reverse T-cell exhaustion by enhancing the level of T-cell cytokines associated with and mediating T-cell activation, enhancing T-cell proliferation and reducing T-cell

apoptosis and downregulating Foxp1. Our study expands the role of miR-149-3p in directly and indirectly modulating T-cell exhaustion and regulating antitumour immunity.

# 5. Methods and materials

## 5.1. Mouse breast tumour cells

4T1 mouse breast cancer cells were obtained from ATCC (Manassas, VA, USA) and cultured in RPMI-1640 full medium (Gibco, Life Technologies, Burlington, Ontario, Canada) with 10% fetal bovine serum (FBS, Gibco), 100 U ml$^{-1}$ of penicillin (Gibco) and 100 μg ml$^{-1}$ streptomycin (Gibco) at 37°C in 5% CO$_2$. A total of $5 \times 10^5$ 4T1 cells were resuspended in 100 μl PBS (Gibco) and injected subcutaneously into the right flank of 6- to 8-week-old female BALB/c mice (Charles River, Saint-Constant, Canada). All tumour-bearing mice were humanely euthanized by CO$_2$ inhalation 16–18 days after tumour cell injection (and before tumours reached 2 cm$^3$) and spleens were collected.

## 5.2. CD8+ T-cell isolation and CD8+PD-1+ and CD8+PD-1− cell sorting

The collected tumour-bearing mice spleens were disaggregated with the flat end of a syringe in 5 ml of RPMI 1640

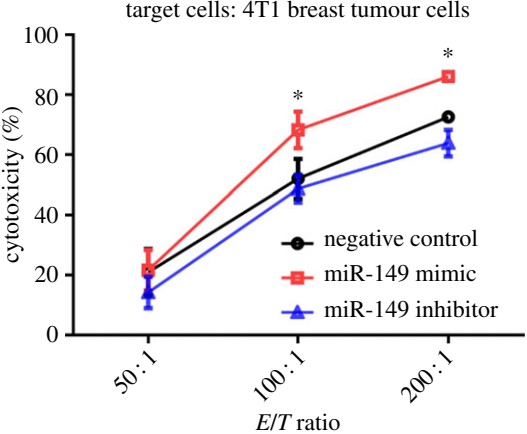

royalsocietypublishing.org/journal/rsob    *Open Biol.* **9**: 190061

**Figure 8.** miR-149-3p treatment improved antitumour cell killing by cytotoxic CD8⁺ T cells *in vitro*. CD8⁺ T cells from tumour-bearing mice were co-cultured with 4T1 cells and treated with miR-149-3p mimic or miR-149-3p inhibitor. A CytoTox 96 non-radioactive cytotoxicity assay to determine 4T1 tumour cell death induced by T cells was performed. CD8⁺ T cells were purified from the collected splenocytes using Miltenyi magnetically labelled beads after transfection. CD8⁺ T cells from tumour-bearing mice spleens were co-cultured for 4 h with 4T1 cells at a ratio of 1 : 50, 1 : 100 and 1 : 200. Data are representative of three independent experiments. Two-way ANOVA analyses were performed to determine statistical significance (*$p < 0.05$).

medium in a 100 mm tissue culture dish. Dispersed cells were filtered through a 40 µm Falcon Cell Strainer (VWR, Mississauga, Ontario, Canada) and CD8⁺ T cells were purified from the collected splenocytes using Miltenyi magnetically labelled beads (Miltenyi Biotec, USA) according to the manufacturer's protocol. CD8⁺PD-1+ and CD8⁺PD-1− cells were separated by fluorescence-activated cell sorting using a Becton Dickinson Aria III FACS (BD Biosciences, Mississauga, Ontario, Canada). For sorting purposes, CD8⁺ T cells were stained with 0.2 µg FITC anti-mouse CD8−, PerCP-eFluor 710 anti-mouse PD1, PerCP-eFluor 710 rat IgG2b Isotype Control or 0.1 µg of Fixable Viability Dye eFluor 506 (eBioscience).

## 5.3. RNA extraction and miRNA cDNA synthesis

An miRNeasy Mini Kit (Qiagen, Toronto, Ontario, Canada) was used to extract RNA from CD8⁺ T cells. The RNA concentration and purity were measured using a NanoDrop ND1000 (Thermo Scientific, Ottawa, Ontario, Canada). Reverse transcription of 500 ng RNA, to generate cDNA, was performed using a miScript II Reverse Transcription kit (Qiagen) according to the manufacturer's instructions.

## 5.4. miRNA geneChip analysis

A total of 100 ng miRNA from each aliquot of CD8⁺PD-1+ and CD8⁺PD-1− T cells were used for Affymetrix GeneChip 3.0 miRNA Array analysis (Affymetrix, Santa Clara, CA, USA). All sample labelling and GeneChip processing was performed at the London Regional Genomics Centre (Robarts Research Institute, London, Ontario, Canada). For analysis, miRNAs were labelled with biotin using a FlashTag Biotin HSR RNA Labeling Kit (Genisphere) according to the manufacturer's instructions. Array data were scanned with the GeneChip Scanner 3000 7G (Affymetrix) and analysed using Affymetrix

GeneChip Command Console software (Affymetrix) by the Partek Genomics Suite (Partek, St Louis, MO, USA).

## 5.5. miRNA reverse transcription–qPCR

miRNA RT–qPCR was performed to verify miRNA levels estimated by expression array analysis. Primer for PCR of selected miR-149-3p was purchased from Qiagen (Qiagen, MS00069222). Snord 6.1 (small nucleolar RNA, C/D box 61) (Qiagen, MS00033705) for mouse was used as a reference miRNA. The qPCR reaction was conducted according to the manufacturer's instructions.

## 5.6. Prediction of miRNA-targeting IRs

miRWalk 2.0 (http://zmf.umm.uni-heidelberg.de/apps/zmf/mirwalk2/generetsys-self.html), as a comprehensive archive gathering 13 prediction datasets including miRWalk, Targetscan, miRanda and others, was used to access the binding site of candidate miRNAs [46]. miRNAs with sequences predicted to bind 3′UTRs of target mRNAs encoding Foxp1, PD-1, TIM-3 and BTLA genes were selected.

## 5.7. Gene expression analysis of PD-1, TIM-3, BTLA and Foxp1

A sensiFASTTM Probe NO-ROX Kit (Bioline, USA) was used to conduct RT–qPCR reactions. Primers of murine PD-1, TIM-3, BTLA, Foxp1 and internal control gene β-actin were PD-1: 5′-GGCCGCCTTCTGTAATGGTTTGA-3′ (forward) and 5′-AGGGGCTGGGATATCTTGTTGAGG-3′ (reverse); TIM-3: 5′-AGTGGGAGTCTCTGCTGGGTTGA-3′ (forward) and 5′-AGGATGGCTGCTGGCTGTTGA-3′ (reverse); BTLA: 5′-GTGAATAAAGAGGCCTTACT-3′ (forward) and 5′-CCTGAACAAGCTTAACTAGA-3′ (reverse); Foxp1: 5′-GCTTCTGCTGACTCTCCTGG-3′ (forward) and 5′-GGAGCCCTTTAGGCTAGCAG-3′ (reverse); β-actin: 5′-AGGGAAATCGTGCGTGACAT-3′ (forward) and 5′-AACCGCTCGTTGCCAATAGT-3′ (reverse). RT–PCR reaction conditions were 95°C for 2 min, followed by 40 cycles at 95°C for 10 s, 58° C for 10 s, then 95°C for 10 s. The qPCR results were analysed using the 2-ΔΔCt method [47].

## 5.8. Mixed lymphocyte reaction

MLRs were performed to detect the cytotoxicity mediated by T cells. CD8⁺ T cells (Miltenyi Biotec) isolated from spleens from BALB/c tumour-bearing mice were co-cultured with C57BL/6 bone marrow-derived DCs. DCs were isolated from tumour-naive 6- to 8-week-old C57BL/6 mice fibulae and tibiae cultured in RPMI-1640 medium containing 10% FBS, 50 ng ml⁻¹ recombinant mouse granulocyte-macrophage colony-stimulating factor (Peprotech, USA) and 50 ng ml⁻¹ recombinant mouse IL-4 (Peprotech, Rocky Hill, USA) for 6 days. CD8⁺ T cells and DCs (10 : 1 ratio, $5 \times 10^5$ T cells and $5 \times 10^4$ DCs per well) were co-cultured in 12-well plates for 48 h and then CD8⁺ T cells were analysed for IR expression, cytokine production and proliferation by flow cytometry.

## 5.9. Flow cytometric analysis

Flow cytometry of CD8⁺ T cells was performed using a Cyto-FLEXS FACS (Beckman Coulter Life Sciences, Mississauga,

Ontario, Canada) to detect IRs, cytokine levels and proliferation. Lymphocytes were stained with 0.2 µg of each of the following MAbs: FITC anti-mouse CD8, PerCP-eFluor 710 anti-mouse PD-1, PE-CY7 anti-mouse TIM-3, APC anti-mouse BTLA, PerCP-eFluor 710 Rat IgG2b Isotype Control, PE Mouse IgG2a κ Isotype Control, FITC Mouse IgG2a κ Isotype Control, PE-CY7 Mouse IgG2a κ Isotype Control, APC Mouse IgG1 κ Isotype Control, PE anti-mouse IL-2 PE anti-mouse TNF-α and PE anti-mouse IFN-γ- (eBioscience, San Diego, CA, USA). A Fixation/Permeabilization Solution Kit (BD Biosciences, USA) was used for intracellular staining where required for different experiments.

## 5.10. T-cell transfection with miR-149-3p mimics or inhibitors

Lymphocytes from 4T1-bearing BALB/c mice spleens were plated with $5 \times 10^6$ cells per well in a 12-well-plate with 900 µl RPMI1640 culture medium plus 10% FBS. A total of 100 µl transfection reagent, containing miR-149-3p mimic (Qiagen, no. MSY0016990) or miR-149-3p inhibitor (Qiagen, no. MIN0016990; controls, no. SI03650318 and no. 1027281, respectively) in 50 µl Opti-MEM and 1 µl Endofectin Max (GeneCopoeia, USA) in 50 µl Opti-MEM, were added after incubation for 30 min. CD8$^+$ T cells were isolated using CD8 MACS Microbeads (Miltenyi Biotec) after transfection for 48 h and were used to further investigate the function of miR-149-3p in regulating PD-1, TIM-3, BTLA and Foxp1 expression. RT–qPCR was performed to assess the expression of PD-1, TIM-3, BTLA and Foxp1.

## 5.11. T-cell apoptosis assays

CD8$^+$ T cells were isolated (Miltenyi Biotec) and washed twice with PBS, and centrifuged at 1500 r.p.m. for 5 min to remove supernatant. The apoptotic cell population was examined using a FITC Annexin V Apoptosis Detection Kit I (BD Pharmingen, CA, USA). T cells were suspended in 200 µl binding buffer and stained with 5 µl Annexin V and PI at

room temperature for 15 min in the dark. The T-cell population was analysed by CytoFLEXS flow cytometry according to the instructions governing the use of the BD Apoptosis Detection Kit. CD8$^+$ T cells were isolated (Miltenyi Biotec) and washed twice with PBS, and centrifuged at 1500 r.p.m. for 5 min to remove supernatant. The apoptotic cell population was examined using a FITC Annexin V Apoptosis Detection Kit I (BD Pharmingen, CA). T cells were suspended in the 200 µl binding buffer and stained with 5 µl Annexin V and PI at room temperature for 15 min in the dark. The T-cell population was analysed by CytoFLEXS flow cytometry according to the instructions governing the use of the BD Apoptosis Detection Kit.

## 5.12. CD8$^+$ T-cell proliferation

CD8$^+$ T cells were labelled with CFSE dye (carboxyfluorescein succinimidyl ester, 1 µl ml$^{-1}$, eBioscience, CA, USA) at 37°C for 15 min and RPMI-1640 with 10% FBS (5 min) was added to terminate the reaction. Antigens (freeze–thawed 4T1 tumour lysate) were then added at a concentration of 50 µg ml$^{-1}$ to simulate T-cell proliferation. CD8$^+$ T cells were collected on day 3 following the addition of lysate and the proliferation rate was assessed by the progressive dilution of CFSE dye. CD8 and IgG2a K on CFSE-labelled T cells were stained using PE anti-mouse CD8 (BioLegend, CA, USA) and PE Mouse IgG2a K Isotype Control antibodies before flow cytometry using CytoFLEXS.

## 5.13. Specific cytotoxic T-lymphocyte response assay

CD8$^+$ T cells from tumour-bearing mice spleens were co-cultured for 4 h with 4T1 cells at a ratio of 1 : 50, 1 : 100 and 1 : 200. Supernatants were then collected to detect lactate dehydrogenase released from T-cell-killed 4T1 tumour cells (CTL response) using a CytoTox 96 Non-Radioactive Cytotoxicity Assay (Promega, CA, USA). The wavelength was set to 450 nm and the absorbance value was measured in different reactions according to the manufacturer's instructions. The percentage of specific cytotoxicity was calculated as follows:

$$\% \text{ cytotoxicity} = \frac{\text{experimental} - \text{effector spontaneous} - \text{target spontaneous}}{\text{target maximum} - \text{target spontaneous}} \times 100\%.$$

## 5.14. Statistical analysis

Two-tailed unpaired Student $t$-tests, one-way ANOVA or two-way ANOVA were applied to determine statistical significance. Significance values are indicated as *($p < 0.05$), **($p < 0.01$).

Data accessibility. This article has no additional data.
Authors' contributions. W.M. conceived and designed the experiments; M.Z., Q.Y., D.G., Y.S., Y.W., R.J., D.L. and Q.L. performed the experiments; M.Z., D.G., Y.Z. and H.W. analysed the data; W.M. and Z.-X.Z. contributed reagents/materials/analysis tools; and M.Z., D.G., W.M., F.A. and J.K. wrote the paper.
Competing interests. The authors declare no conflict of interest.
Funding. This study was supported by grants from the National Natural Science Foundation of China (grant nos 81673009 and 81660274). F.A. is supported by a King Abdullah Scholarship.
Ethics. Animals were maintained under conventional conditions in the Animal Care Facility (Western University) in accord with the guidelines established by Canadian Council on Animal Care (CCAC).

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
