## [Reviewer comments · Open Biology]

Review History

RSOB-19-0061.R0 (Original submission)

Review form: Reviewer 1

Recommendation

Major revision is needed (please make suggestions in comments)

Are each of the following suitable for general readers?

- a) **Title**
No

- b) **Summary**
No

c) **Introduction**
Yes

Is the length of the paper justified?

Yes

Should the paper be seen by a specialist statistical reviewer?

No

Is it clear how to make all supporting data available?

No

Is the supplementary material necessary; and if so is it adequate and clear?

Yes

Do you have any ethical concerns with this paper?

No

Comments to the Author

The manuscript entitled "miR-149-3p reverses CD8 T cell exhaustion by reducing inhibitory receptors and promoting cytokine secretion in breast cancer cells" by Zhang et al. discusses the role of miRNA in regulating the expression of critical proteins involved in T cell exhaustion. There is need to study this concept due to its significance in the field. However, the present study is not well designed and lacks scientific rigor. The experiments were performed in splenic CD8 T cells but the authors emphasized on breast cancer cells/tumor tissues in the title, abstract, introduction and discussion. No experiments were performed to show how tumor-infiltrating CD8 T cells are impacted. Experimental design is not very rigorous to convincingly show that the observation of in vitro experiments are solely due to miRNA over-expression. Specifically, the transfection of mixed lymphocyte population with miRNA and then sorting of CD8 T cells is not appropriate. A lot of cell death that can occur in other cells types can activate or alter phenotype of CD8 T cells leading to misleading observations.

Comments:

1. The fact that no data on T cells were provided from tumor tissues but authors consistently mentioned impact of T cells in cancer cells is scientifically inaccurate. For instance in section 2.3 authors state "Proliferation of CD8+ T cells declined in tumor-bearing mice on day 3" but the figure legend is states "CD8+ T cells isolated from spleens of tumor-bearing mice proliferated less and were more apoptotic". This inaccuracy is a major concern.
2. In Figure 2A;B- It is apparent that total cell counts in naive and tumor-bearing spleens are different. Authors should have shown percentages of CD8 T cells in both the populations prior to performing these experiments. It is likely that CD8 count would be different to begin with.
3. MiRNA microarray data is not well discussed. There is no explanation of how data is analyzed and what are the statistical tests for the cut-off for selecting the differentially expressed miRNAs. The rationale of picking four of the downregulated miRNAs is not clear.
4. Experimental design of transfecting lymphocytes with as a whole and the sorting CD8 population to study the impact of miRNA is wrong. These CD8 cells are very sensitive to changes in other cells behavior and transfection process itself can cause lot of phenotypic and functional changes in other cells that may impair CD8 cells ability to respond appropriately. Why authors did not start with sorted CD8 T cells to begin with the transfection?

5. To convincingly show that miRNA directly regulate the changes in target mRNA levels, authors should have performed the reported assays using the 3'UTR cloning. This is a standard method to validate the miRNA-target interaction. From the current data, although it is clear that miRNA-149-3p is impacting target protein levels but this could be due to other mechanism besides direct miRNA interaction. To prove this, miRNA-target interaction data should be shown.
6. In Figure 6; the cellular changes in cytokine levels are not very clear. The flow data is not convincing and does not match well with the corresponding histograms.
7. Figure legends are not self-explanatory and are also not well described elsewhere. It will be difficult for readers to understand the experimental context of each panel in the figure.
8. Figure labels do not match with figure legends and or main text. There are sub-panels in Figure 2, 3 which are not mentioned in the legend or main text.
9. Numerous syntax and typo errors throughout the manuscript should be corrected.

Decision letter (RSOB-19-0061.R0)

10-May-2019

Dear Dr Min,

We are writing to inform you that the Editor has reached a decision on your manuscript RSOB-19-0061 entitled "miR-149-3p reverses CD8 T cell exhaustion by reducing inhibitory receptors and promoting cytokine secretion in breast cancer cells", submitted to Open Biology.

As you will see from the reviewer's comments below, there are a number of criticisms that prevent us from accepting your manuscript at this stage. The reviewer suggests, however, that a revised version could be acceptable, if you are able to address their concerns. If you think that you can deal satisfactorily with the reviewer's suggestions, we would be pleased to consider a revised manuscript.

The revision will be re-reviewed, where possible, by the original referees. As such, please submit the revised version of your manuscript within four weeks. If you do not think you will be able to meet this date please let us know immediately.

When submitting your revised manuscript, please respond to the comments made by the referee(s) and upload a file "Response to Referees" in "Section 6 - File Upload". You can use this to document any changes you make to the original manuscript. In order to expedite the

processing of the revised manuscript, please be as specific as possible in your response to the referee(s).

Please see our detailed instructions for revision requirements
<https://royalsociety.org/journals/authors/author-guidelines/>

Sincerely,

The Open Biology Team
mailto:openbiology@royalsociety.org

Reviewer's Comments to Author(s):

Referee:

Comments to the Author(s)

The manuscript entitled "miR-149-3p reverses CD8 T cell exhaustion by reducing inhibitory receptors and promoting cytokine secretion in breast cancer cells" by Zhang et al. discusses the role of miRNA in regulating the expression of critical proteins involved in T cell exhaustion. There is need to study this concept due to its significance in the field. However, the present study is not well designed and lacks scientific rigor. The experiments were performed in splenic CD8 T cells but the authors emphasized on breast cancer cells/tumor tissues in the title, abstract, introduction and discussion. No experiments were performed to show how tumor-infiltrating CD8 T cells are impacted. Experimental design is not very rigorous to convincingly show that the observation of in vitro experiments are solely due to miRNA over-expression. Specifically, the transfection of mixed lymphocyte population with miRNA and then sorting of CD8 T cells is not appropriate. A lot of cell death that can occur in other cells types can activate or alter phenotype of CD8 T cells leading to misleading observations.

Comments:

1. The fact that no data on T cells were provided from tumor tissues but authors consistently mentioned impact of T cells in cancer cells is scientifically inaccurate. For instance in section 2.3 authors state "Proliferation of CD8+ T cells declined in tumor-bearing mice on day 3" but the figure legend is states "CD8+ T cells isolated from spleens of tumor-bearing mice proliferated less and were more apoptotic". This inaccuracy is a major concern.
2. In Figure 2A;B- It is apparent that total cell counts in naive and tumor-bearing spleens are different. Authors should have shown percentages of CD8 T cells in both the populations prior to performing these experiments. It is likely that CD8 count would be different to begin with.
3. MiRNA microarray data is not well discussed. There is no explanation of how data is analyzed and what are the statistical tests for the cut-off for selecting the differentially expressed miRNAs. The rationale of picking four of the downregulated miRNAs is not clear.
4. Experimental design of transfecting lymphocytes with as a whole and the sorting CD8 population to study the impact of miRNA is wrong. These CD8 cells are very sensitive to changes in other cells behavior and transfection process itself can cause lot of phenotypic and functional changes in other cells that may impair CD8 cells ability to respond appropriately. Why authors did not start with sorted CD8 T cells to begin with the transfection?

5. To convincingly show that miRNA directly regulate the changes in target mRNA levels, authors should have performed the reported assays using the 3'UTR cloning. This is a standard method to validate the miRNA-target interaction. From the current data, although it is clear that miRNA-149-3p is impacting target protein levels but this could be due to other mechanism besides direct miRNA interaction. To prove this, miRNA-target interaction data should be shown.

6. In Figure 6; the cellular changes in cytokine levels are not very clear. The flow data is not convincing and does not match well with the corresponding histograms.

7. Figure legends are not self-explanatory and are also not well described elsewhere. It will be difficult for readers to understand the experimental context of each panel in the figure.

8. Figure labels do not match with figure legends and or main text. There are sub-panels in Figure 2, 3 which are not mentioned in the legend or main text.

9. Numerous syntax and typo errors throughout the manuscript should be corrected.

Author's Response to Decision Letter for (RSOB-19-0061.R0)

See Appendix A.

RSOB-19-0061.R1 (Revision)

Review form: Reviewer 1

Recommendation

Major revision is needed (please make suggestions in comments)

Are each of the following suitable for general readers?

- a) **Title**
Yes
- b) **Summary**
Yes
- c) **Introduction**
Yes

Is the length of the paper justified?

Yes

Should the paper be seen by a specialist statistical reviewer?

No

Is it clear how to make all supporting data available?

Yes

Is the supplementary material necessary; and if so is it adequate and clear?

Yes

Do you have any ethical concerns with this paper?

No

Comments to the Author

Authors have provided explanation to various comments and have clarified most of the concerns. However, I still am not convinced with the fact that experiment (Figure 5), a major finding and the title of the study is not properly designed. The response by the authors justified my point that it is not properly performed and acknowledged the CD8 T cell "exhaustion" in the experimental setup. I would recommend them to show data only on CD8 T cells instead of using whole splenocyte populations. I will not be convinced with the data obtained from the current experimental approach. Additionally, there is not data to show the transfection efficiency in CD8 T cells in their experiment. I would highly encourage the authors to perform this experiment with right control to demonstrate the significance of miR-149-3p in CD8 T cell exhaustion.

Review form: Reviewer 2

Recommendation

Accept as is

Are each of the following suitable for general readers?

- a) **Title**
Yes
- b) **Summary**
Yes
- c) **Introduction**
Yes

Is the length of the paper justified?

Yes

Should the paper be seen by a specialist statistical reviewer?

No

Is it clear how to make all supporting data available?

Not Applicable

Is the supplementary material necessary; and if so is it adequate and clear?

Not Applicable

Do you have any ethical concerns with this paper?

No

Comments to the Author

In the manuscript entitled „miR-149-3p reverses CD8 T cell exhaustion by reducing inhibitory receptors and promoting cytokine secretion in breast cancer cells” Zhang and coworkers have investigated the regulation of the expression of checkpoint inhibitors in a murine model of breast cancer. The results presented in this work indicate that miR-149-3p regulates the expression of PD-1, TIM-3, BTLA, and Foxp1. More importantly, the authors have shown that miR149-3p mimic reverses the exhaustion of CD8+ T cells.

Despite the fact that the study is limited to a murine model of breast cancer and that in some cases the effects of the miRNA mimic are modest, overall the results presented by the authors are very interesting. Moreover the paper is well written, the results are presented in a very logical way and the conclusions supported by the data. Therefore, I recommend this manuscript for publication in Open Biology without any further request.

Decision letter (RSOB-19-0061.R1)

21-Jun-2019

Dear Dr Min,

We are writing to inform you that the Editor has reached a decision on your manuscript RSOB-19-0061.R1 entitled "miR-149-3p reverses CD8 T cell exhaustion by reducing inhibitory receptors and promoting cytokine secretion in breast cancer cells", submitted to Open Biology.

As you will see from the reviewers' comments below, there are a number of criticisms that prevent us from accepting your manuscript at this stage. The reviewers suggest, however, that a revised version could be acceptable, if you are able to address their concerns. If you think that you can deal satisfactorily with the reviewer's suggestions, we would be pleased to consider a revised manuscript.

The revision will be re-reviewed, where possible, by the original referees. As such, please submit the revised version of your manuscript within four weeks. If you do not think you will be able to meet this date please let us know immediately.

When submitting your revised manuscript, please respond to the comments made by the referee(s) and upload a file "Response to Referees" in "Section 6 - File Upload". You can use this to document any changes you make to the original manuscript. In order to expedite the processing of the revised manuscript, please be as specific as possible in your response to the referee(s).

Please see our detailed instructions for revision requirements. It is essential these instructions are followed carefully to minimize any delay to publication:
<https://royalsociety.org/journals/authors/author-guidelines/>

Sincerely,
The Open Biology Team
mailto: openbiology@royalsociety.org

Reviewer(s)' Comments to Author(s):

Referee: 1

Comments to the Author(s)

Authors have provided explanation to various comments and have clarified most of the concerns. However, I still am not convinced with the fact that experiment (Figure 5), a major finding and the title of the study is not properly designed. The response by the authors justified my point that it is not properly performed and acknowledged the CD8 T cell "exhaustion" in the experimental setup. I would recommend them to show data only on CD8 T cells instead of using whole splenocyte populations. I will not be convinced with the data obtained from the current experimental approach.

Additionally, there is not data to show the transfection efficiency in CD8 T cells in their experiment. I would highly encourage the authors to perform this experiment with right control to demonstrate the significance of miR-149-3p in CD8 T cell exhaustion.

Referee: 2

Comments to the Author(s)

In the manuscript entitled „miR-149-3p reverses CD8 T cell exhaustion by reducing inhibitory receptors and promoting cytokine secretion in breast cancer cells” Zhang and coworkers have investigated the regulation of the expression of checkpoint inhibitors in a murine model of breast cancer. The results presented in this work indicate that miR-149-3p regulates the expression of PD-1, TIM-3, BTLA, and Foxp1. More importantly, the authors have shown that miR149-3p mimic reverses the exhaustion of CD8+ T cells.

Despite the fact that the study is limited to a murine model of breast cancer and that in some cases the effects of the miRNA mimic are modest, overall the results presented by the authors are very interesting. Moreover the paper is well written, the results are presented in a very logical way and the conclusions supported by the data. Therefore, I recommend this manuscript for publication in Open Biology without any further request.

Author's Response to Decision Letter for (RSOB-19-0061.R1)

See Appendix B.

RSOB-19-0061.R2 (Revision)

Review form: Reviewer 1

Recommendation

Accept as is

Do you have any ethical concerns with this paper?

No

Comments to the Author

Manuscript is acceptable now.

Decision letter (RSOB-19-0061.R2)

17-Sep-2019

Dear Dr Min

We are pleased to inform you that your manuscript entitled "miR-149-3p reverses CD8 T cell exhaustion by reducing inhibitory receptors and promoting cytokine secretion in breast cancer cells" has been accepted by the Editor for publication in Open Biology.

If applicable, please find the referee comments below. No further changes are recommended.

Sincerely,

The Open Biology Team
mailto: openbiology@royalsociety.org

Appendix A

Point-to-point response to reviewer's comments

1. The fact that no data on T cells were provided from tumor tissues but authors consistently mentioned impact of T cells in cancer cells is scientifically inaccurate. For instance in section 2.3 authors state 'Proliferation of CD8+ T cells declined in tumor-bearing mice on day 3' but the figure legend is states "CD8+ T cells isolated from spleens of tumor-bearing mice proliferated less and were more apoptotic". This inaccuracy is a major concern.

Response: We agree these confused statements, and we have replaced following sentences to clarify the data: "Proliferation of CD8+ T cells was declined from tumor-bearing mice spleen on day 3". These revised statements are now shown in the section of 2.3, in page 6.

2. In Figure 2A;B- It is apparent that total cell counts in naive and tumor-bearing spleens are different. Authors should have shown percentages of CD8 T cells in both the populations prior to performing these experiments. It is likely that CD8 count would be different to begin with.

Response: In light of reviewer's suggestion, we added the ratio of splenic CD8+ T cells in the naive mice and tumor-bearing mice. The new data are shown in Supplementary Figure 1. We have also added description of these new data in the section of Result 2.1 in page 4. We also accordingly modified the figure legend of Figure 2.

Supplementary Figure 1

Supplementary Figure S1. Percentages of splenic CD8+ T cells in naive and tumor-bearing mice (A, B)

Detection of CD8+ T cells by flow cytometry. Spleen cells were collected from naive mice and tumor-bearing mice on day 16-18 after tumor cell injection. Flow cytometry was performed after FITC anti-mouse CD8 staining. Data are representative of three independent experiments. Unpaired Student *t*-tests were performed to determine statistical significance (** $p < 0.01$).

3. miRNA microarray data is not well discussed. There is no explanation of how data is analyzed and what are the statistical tests for the cut-off for selecting the differentially expressed miRNAs. The rationale of picking four of the downregulated miRNAs is not clear.

Response: As advised, we have added a paragraph to describe the data analysis and statistical tests of microarray data for the selection of miRNA. These new sentences are shown in the section of 2.4, page 7 and section of 6.4, page 18. The cut-off for selecting candidate miRNAs was determined as a fold change of ± 1.5 (PD-1+ vs PD-1-, $p < 0.05$).

4. Experimental design of transfecting lymphocytes with as a whole and the sorting CD8 population to study the impact of miRNA is wrong. These CD8 cells are very sensitive to changes in other cells behavior and transfection process itself can cause lot of phenotypic and functional changes in other cells that may impair CD8 cells ability to respond appropriately. Why authors did not start with sorted CD8 T cells to begin with the transfection?

Response: We indeed conducted the experiments as the reviewer suggested, that is, sorting CD8 cells first then transfecting the sorted cells. Unfortunately, we encountered two problem in these experiments. First, the number of sorted CD8+ T cells are too low to do the subsequent functional experiments. Additionally, the cell transfection as well as following cell culture of CD8+ T cells were difficult after sorting, possibly due to the in vitro-manipulated "exhausting" T cells were more fragile and sensitive to the process of cell sorting. Therefore, we transfected whole T cells after induction of exhaustion, then analyzed the CD8+ T cell phenotype and function through double staining with anti-CD8 and other antibodies. We agree this might impair CD8 T cell response, however, our data demonstrated that miR-149-3p can modulate IRs expression and cytokine secretion in Figure 2.5, 2.6 and Supplementary Figure 2.

5. To convincingly show that miRNA directly regulate the changes in target mRNA levels, authors should have performed the reported assays using the 3'UTR cloning. This is a standard method to validate the miRNA-target interaction. From the current data, although it is clear that miRNA-149-3p is impacting target protein levels but this could be due to other mechanism besides direct miRNA interaction. To prove this, miRNA-target interaction data should be shown.

Response: We agree that the miRNA binding assay using dual-fluorescent assay is a standard method, and will provide more direct evidence to validate the miRNA-target interaction. Unfortunately, the experiments of selecting promoter binding sites, cloning of binding sequences and mutants, making vectors and validating the binding sites, all of these procedures are time consuming, which is not able to be completed within the 4 weeks (even 8 weeks) of revision time period. Nonetheless, although we were not able to demonstrated the direct binding data, the results in this study clearly showed miR-149-3p can reduce the levels of inhibitory receptors in CD8+ exhausted T cells in the section of Results 2.5.

6. In Figure 6; the cellular changes in cytokine levels are not very clear. The flow data is not convincing and does not match well with the corresponding histograms.

Response: In order to verify the cytokine changes in Figure 6, we repeated our experiments and detected the cytokine levels using RT-qPCR. The new data are shown in Supplementary Figure S2. We also described these new data in the section of Results 2.6, page 12. We also modified the figure legend of Figure 6.

Supplementary Figure 2

Supplementary Figure S2. miR-149-3p restored activity-associated cytokine levels in exhausted CD8+ T cells. (A) Detection of cytokine IL-2, TNF- α and IFN- γ levels by RT-qPCR. Spleen cells collected from tumor-bearing mice were respectively transfected with control miRNA, miR-149-3p mimics and miR-149-3p inhibitors for 48 hrs. After transfection, CD8+ T cells were purified from the collected splenocytes using Miltenyi magnetically-labeled beads (Miltenyi Biotec, USA). Detection of cytokine IL-2, TNF- α , and IFN- γ levels on CD8+ T cells by qPCR were performed using CD8+ T cells mRNA. Data are representative of three independent experiments. Unpaired Student's t-tests were performed to determine statistical significance (* $p < 0.05$, ** $p < 0.01$).

7. Figure legends are not self-explanatory and are also not well described elsewhere. It will be difficult for readers to understand the experimental context of each panel in the figure.

Response: As suggested, we have revised all Figure legends.

8. Figure labels do not match with figure legends and or main text. There are sub-panels in Figure 2, 3 which are not mentioned in the legend or main text.

Response: As suggested, we have revised the description of Figure 2 and Figure 3 by integrating the data of sub-panel into the text of Results and Figure legends.

9. Numerous syntax and typo errors throughout the manuscript should be corrected.

Response: We appreciate the reviewer's careful review. We have revised this manuscript and correct all typo errors.

Appendix B

Point-to-point response to reviewer's comments

Referee: 1

1. Authors have provided explanation to various comments and have clarified most of the concerns. However, I still am not convinced with the fact that experiment (Figure 5), a major finding and the title of the study is not properly designed. The response by the authors justified my point that it is not properly performed and acknowledged the CD8 T cell "exhaustion" in the experimental setup. I would recommend them to show data only on CD8 T cells instead of using whole splenocyte populations. I will not be convinced with the data obtained from the current experimental approach.

Response: As advised, we have conducted the experiments in light of reviewer's suggestion. We isolated CD8+ T cells first then detected IRs expression after transfection with miR-149 mimics or inhibitors. We have replaced Figure 5 with the new experimental data, which are included in the Results 2.5 in page 10. We also accordingly modified the figure legend of Figure 5.

Figure 5

Figure 5. Altered expression of markers of exhaustion in T cells after treatment with miR-149-3p mimic or miR-149-3p inhibitor. (A) Detection of mRNAs encoding PD-1, TIM-3, BTLA by RT-

qPCR analysis. CD8+ T cells were purified from the tumor-bearing mice splenocytes using Miltenyi magnetically-labeled beads and then transfected with control miRNA, miR-149-3p mimics or miR-149-3p inhibitors for 48 hrs. After transfection, RT-qPCR was performed to detect PD-1, TIM-3, BTLA and Foxp1 mRNA levels in CD8+ T cells. **(B)** Examination of inhibitory receptors on CD8+ T cells by flow cytometry. CD8+ T cells were purified from tumor-bearing mice splenocytes and transfected with control miRNA, miR-149-3p mimics or miR-149-3p inhibitors for 48 hrs. Flow cytometry was performed after transfection. Data are representative of three independent experiments. One-way ANOVA analyses were performed to determine statistical significance ($*p < 0.05$, $**p < 0.01$).

2. Additionally, there is not data to show the transfection efficiency in CD8 T cells in their experiment. I would highly encourage the authors to perform this experiment with right control to demonstrate the significance of miR-149-3p in CD8 T cell exhaustion.

Response: We indeed conducted the experiments as the reviewer suggested. In light of reviewer's suggestion, we have put transfection efficiency of miR-149-3p data in Supplementary Figure 2. The data are included the Results 2.5 in page 10.

Supplementary Figure 2

Supplementary Figure S2. Detection of miR-149-3p level after transfection. CD8+ T cells were purified from the tumor-bearing mice splenocytes using Miltenyi magnetically-labeled beads and transfected with control miRNA, miR-149-3p mimics, and miR-149-3p inhibitors for 48 hrs. After transfection, miRNeasy Mini Kit and miScript II Reverse Transcriptase Kit were used for miRNA isolation and cDNA synthesis. miR-149-3p levels were detected by RT-qPCR using miScript Primer Assay. Data are representative of three independent experiments. Unpaired Student's t-tests were performed to determine statistical significance ($** p < 0.01$).